# Animal Health Discourse during Ecological Crises in the Media—Lessons Learnt from the Flood in Thessaly from the One Health Perspective

**DOI:** 10.3390/vetsci11040140

**Published:** 2024-03-22

**Authors:** Eleftherios Meletis, Andrzej Jarynowski, Stanisław Maksymowicz, Polychronis Kostoulas, Vitaly Belik

**Affiliations:** 1Faculty of Public and One Health, University of Thessaly, 43100 Karditsa, Greece; elmeletis@uth.gr (E.M.); pkost@uth.gr (P.K.); 2Balkan Association for Vector-Borne Diseases, 21000 Novi Sad, Serbia; 3Epidemic Intelligence Unit, Polish Society of Hygiene, 50-950 Wrocław, Poland; 4System Modeling Group, Institute of Veterinary Epidemiology and Biostatistics, Freie Universität Berlin, 14163 Berlin, Germany; vitaly.belik@fu-berlin.de; 5School of Public Health, Collegium Medicum, University of Warmia and Mazury in Olsztyn, 10-719 Olsztyn, Poland; stanislaw.maksymowicz@uwm.edu.pl

**Keywords:** crisis management, veterinary public health, animal health, perception of animals, One Health

## Abstract

**Simple Summary:**

Climate change and conflicts are making disasters involving animals’ health more common. We studied how the Greek media covered the major flood in Thessaly, September 2023. We looked at how people felt about animal health and disease risks. This study found that crisis plans did not focus enough on animals and on communication. Our research also showed that (i) people rely on each other for health information rather than experts during a crisis, and (ii) media often sensationalize animal deaths. This highlights the need for better crisis management (with the use of social media listening tools) in veterinary education and practices to handle disasters.

**Abstract:**

Due to the increasing risk of extreme events caused by climate change (i.e., floods, fires and hurricanes) or wars, European veterinary public health may need some improvement. Utilizing a mix of qualitative (participatory observation) and quantitative methods (Internet mining), we analyzed the Greek media’s responses to the millennial flood in Thessaly (September 2023), focusing on animal health (including wild, companion animals and livestock) and public sentiment towards epizootic/epidemic threats. The study revealed a gap in crisis management plans regarding veterinary-related issues, emphasizing the need for comprehensive emergency response strategies. Our findings show how (i) the lay referral system is projecting the perception of epidemic threats into the population; (ii) the emotional load of images of animal carcasses is misused by media creators aiming for a big audience; and (iii) pets’ owners are creating online communities for the searching and treatment of their pets. Our results stress the importance of integrating crisis communication in consecutive phases of the discourse, such as the following: (i) weather change; (ii) acute flood; (iii) recovery; and (iv) outbreaks, into veterinary practices to better prepare for such disasters.

## 1. Introduction

During disasters such as floods, livestock, wild and companion animals can face significant challenges and dangers. Crisis management plans mainly concern agricultural and human health issues [1], and veterinary-related issues are less covered. Even though there are guides for veterinary and humanitarian professionals to plan emergency responses for the care and welfare of animals for various topics on disasters, such as principles of disaster management, operation planning and team deployment [2], the European perspective seems to be missing. Our experience from Storm Daniel in September 2023, and other events such as the Oder River disaster [3,4], showed that crisis communication, in particular between different expert groups, authorities and lay people, is of a big concern and must be taken into account. According to official estimates [4], around 250,000 livestock (75,723 sheep and goats, 21,342 pigs, 6709 cattle and 131,795 birds) disappeared in Thessaly during the first wave of flooding (7–12 September 2023). Most of them drowned since animals unable to find higher ground or to safely escape became submerged and drifted away from rising waters. To mitigate the impact of floods on animals also in the One Health context, multiple efforts were taken. This includes providing proper shelter, securing food and water supplies and having sophisticated evacuation plans in combination with intelligent monitoring systems. Over the last few years, Greek farmers widely applied precision farming techniques (farm management using information technology, control systems, sensors and software to monitor, measure and optimize agricultural production processes). This allows for the monitoring of the impact of Storm Daniel on crops and livestock [5]. Furthermore, we analyzed human gastroenterological disease dynamics using real-time syndromic surveillance systems [6].

## 2. Materials and Methods

Our study employs both qualitative and quantitative methods to assess data from Greek digital traditional and so-called “new” media during September 2023. We collected 13,873 mentions [7] related to the flood using Brand24 (the supply of information) and Google Trends (the demand of information). The goal of this analysis is to investigate the social reaction [8] to the threat (bottom-up approach), regarding animal health [9]. For the sake of triangulation, a participatory observation perspective (by EM and AJ) was included.

## 3. Results

The information supply reached approximately the entire national population at the peak week, resulting in almost millions of searches and thousands of mentions in Greece.

### 3.1. Time Series Analysis of Search Queries, Mentions, Reach and Sentiment

Figure 1a shows, for the period when Storm Daniel occurred in Thessaly, Greece, the number of flood-related mentions, sentiment (the percentage of negative mentions) and the related social and traditional media reach. The latter is measured as the percentage of the Greek Internet user population. For example, 100% would mean that on average, every inhabitant of Greece was exposed to flood information at once. Figure 1b describes flood-, animal- and (both human- and animal-) health-related Google queries in relative search volume (RSV). It is evident that a peak occurs during the period of the storm, and most of the news has a “negative” sentiment since they describe the response (“rage”) of the people who (i) were affected by the storm and (ii) were observing the situation via the mentions. There is no clear increase in discussion on animal- and epizootic-related issues in Google Trends (with a slight peak of searches on animal health after the first wave of the flood). Based on the inspection of the sources, health- and animal-related issues (only 1330 mentions being 9.6% of the sample) are secondary to topics such as governmental incompetence, which is similar to other studies [9]. Most of the mentions refer to farmers who not only lost their livestock but also the equipment (building, machinery) and animal feed. In Greece, livestock (sheep, goats, cattle and pigs)-keeping farmers are usually cultivating land too, e.g., corn (animal feed) and cotton (as an extra crop) production. Early October is harvesting time, thus many acres of crops ready to be harvested were destroyed.

### 3.2. Topics and Information Gap

Most of the mentions refer to the extent of the damage and point out the need for the proper management of the situation being of a major public health concern.

There is no clear increase in discussion on animal- and epizootic-related issues in Google Trends, with a slight peak of search queries on animal health after the first wave of the flood.

The observed negative sentiment related to the situation can be attributed to the fact that Greece, since 2012, is experiencing a financial crisis. To this end, many discussions about farmers’ income and tax happened. Additionally, public authorities also related to insurance organizations do not have the ability to intervene and compensate the losses of the farmers (a long tail of negative emotions after the flood).

We observed an information gap between veterinary epidemiologists’ opinions and the interest among the public (Table 1) for various visible ailments. This gap might be explained by the concept describing the behavior of patients, known as the lay referral system [10]. It is the process by which people seek health care advice and treatment from non-professional sources, such as family, friends or community members—whom they trust—as opposed to a professional system in which the trust is limited [10]. Patients tend to be guided by their own subjective perception of ailments. It is similar to the perception of epidemic threats, which could be subdivided into imaginable and visible threats. The latter might replace (cognitive dissonance) threats that are difficult to imagine. The lay referral system is a network of informal relationships that people use to obtain health care information and advice. For this reason, we see that it is particularly important to properly inform citizens about actual threats, thus eliminating the impact of mistakes generated by the lay referral system. It may also be useful to use networks of non-professional actors (e.g., by creating local community-based information centers) to disseminate accurate information.

### 3.3. Phases of Animal and Disease Discourse

The theory and practice of crisis management involving animals typically include the following phases: (i) pre-catastrophe, (ii) emergency response, (iii) information dissemination, (iv) post-catastrophe infectious disease outbreaks, (v) recovery and rebuilding, and (vi) reflection and lessons learned [2,3,8,11].

We can empirically distinguish the following phases [11]: (i) Storm Daniel phase (discussion on weather, little concern about animals and diseases (Figure 1 and Figure 2, green)); (ii) acute flood phase (when animal rescue actions were going on with a massive flow of information without the emotional load (Figure 1a, pink)), (iii) recovery phase (when carcasses were removed and losses calculated (Figure 1a and Figure 2, blue)), and (iv) flood-related outbreaks phase with the peak on 21.09 related to animal diseases after feeding at flooded areas (Figure 1b and Figure 2, purple).

## 4. Discussion

Due to climate change [12] and anthropogenic impacts [13], Europe is expecting more and more threats for One Health (especially for veterinary issues) in the future. We demonstrated where information gaps may appear between veterinarians and the public (e.g., concerning complex associations between flood and diseases). The change in perceptions of the environmental crises through climate change is observable globally, but locally, further research using the social science apparatus is needed [14,15] to understand local vulnerabilities and resilience for building good early warning systems [16].

Our results regarding topics, phases and the information gap are substantial, and in the context of veterinary science, entirely new for Europe. In the wake of disasters like floods, fires and hurricanes, it becomes evident that livestock and companion animals face considerable challenges. While there are guides for veterinary and aid workers on disaster management and animal welfare (i.e., [2]), a European perspective, as evidenced in the Thessaly flood response, is conspicuously absent. This study highlights three crucial findings and lessons learnt for better preparation during such events [17]:The influence of the lay referral system in projecting perceptions of epidemic threats (public versus experts in epizootiology). During the analysis, it turned out that in crisis management, it may be useful to use networks of non-professional actors (e.g., farmers or other animal owners)—by creating local community-based information centers, i.e., veterinarians [18]—to disseminate accurate information and fill the information gap in almost real-time.The media influence on shaping discourse (e.g., sensationalizing reports on animal carcasses and neglecting other topics). The professionals must be aware that their intensive work in the movement/treatment of livestock will be ‘invisible’ for the general population, and the mainstream media will select message topics according to their reach potential, which is not always the same as their relevance.The formation of ad hoc online communities or networks (i.e., networks of pet owners for search and treatment efforts). Overall, this should be assessed positively, as it creates networks [19] of lay support (among farmers and pet owners), but it is also likely that these networks will be able to spread misinformation or disinformation.

These insights underscore the necessity for comprehensive emergency response strategies that include animal care.

Our analysis has all the limitations of an ecological type of study design (a type of observational study that examines the relationships between variables with data at the population or group level rather than at the individual level). Additionally, proprietary algorithms involved in the data selection for both mentions and searches may be biased. Moreover, an increased incidence in Summer/Autumn 2023 of vector-borne diseases such as West Nile fever [20] particularly in Thessaly was discussed after the investigated period.

## 5. Conclusions

To mitigate the impacts of disasters on wild, livestock and companion animals, it is essential for the authorities in charge to not only have emergency plans in place [21] but also to possess crisis communication skills [22] and action plans for the local communities in this matter (Figure 3). There is a global change in the perception of crises (which we are likely to experience increasingly due to climate change [14,15]), but more research using the social sciences is needed locally to understand local vulnerabilities and resilience. This includes early warning systems or media monitoring activity during a catastrophic event. Our flood case study underscores the pivotal importance of crisis management in shaping public responses to veterinary public health measures in the so-called two-way communication [22]. These evaluations bolster the case for adopting best practices in risk communication [8], which include integrating risk communication strategies into policy formulation and accommodating media requirements for proactive planning before events (see PRE-CRISIS (Figure 3)). Veterinary officers must be aware of what may drive the interest of the public in their regions. The perception of diarrhea in humans or animals will be more important than the introduction of (re-) emerging or long-term contamination, thus the mismatch between priorities of public health specialists and lay people will be different [9]. Moreover, at least in Thessaly (which does not need to be the case in other settings [7]), the bias of the interest in companion animals in the social/local media may suggest whose voice may be the loudest in the next catastrophe. Thus, mapping the typical needs of the local population will be crucial for an adequate response, and it cannot be conducted without setting up a media monitoring system (other sociological methods, such as surveys or individual/group interviews, may have only supplementary roles). In the CRISIS CUMULATION phase, a response to community apprehensions and feedback from animal owners/breeders about where and how to safeguard or displace is needed through social and local media. From social media listening, the places of animals in need can be detected, and this information may be an additional layer to conventional channels [2]. Quick animal displacement and safeguarding advice should be given to residents of affected regions (see the Australian example [21]). In the LATE CRISIS phase, conveying messages with care and understanding, maintaining and recognizing uncertainties and offering practical advice for Internet communities to follow are crucial (as in our case, the supplementation of vitamins among dogs that have eaten dead fish). Social media listening would show information about current infectious outbreaks, and veterinarians could not only quickly detect problems [11] but could also provide some veterinary advice through social media (some equivalent to distance diagnostics and treatment in crisis situations).

Because local veterinary authorities, animal welfare organizations and animal breeders may play a crucial role in rescue and relief efforts during disaster events, the aforementioned issues should be included in the veterinarian curriculum.

## Figures and Tables

**Figure 1 vetsci-11-00140-f001:**
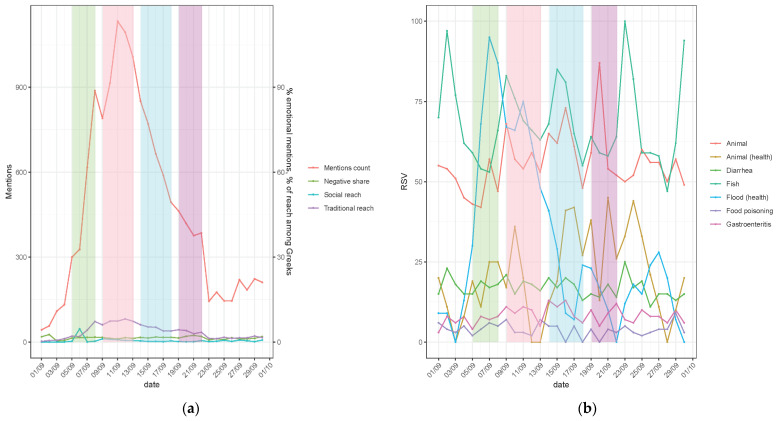
(**a**) The flood in traditional and social media daily (extracted using Brand24). Daily count of mentions and their reach. (**b**) Google search queries as normalized daily values (relative search volume, RSV). Colors represent the following phases: (i) weather change and storms (green); (ii) acute flood (pink); (iii) recovery after main flood (blue); and (iv) disease outbreaks (purple).

**Figure 2 vetsci-11-00140-f002:**
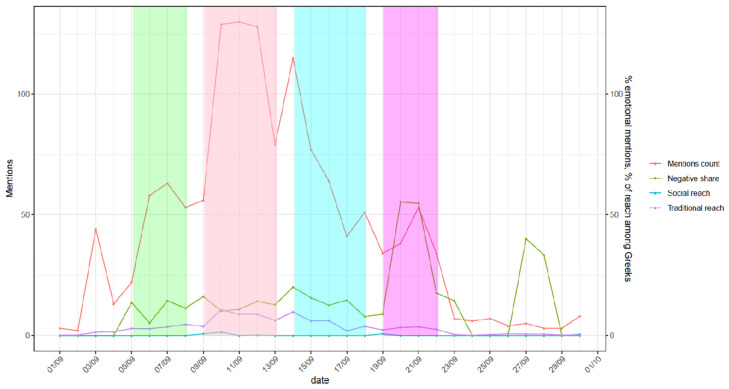
Interest on the Internet for human and animal health (i.e., diarrhea) related to the flood as measured by mentions count (left axis), social and traditional reach (right axis) and the corresponding sentiment of disease-related ideas (right axis). Color code for phases as in Figure 1.

**Figure 3 vetsci-11-00140-f003:**
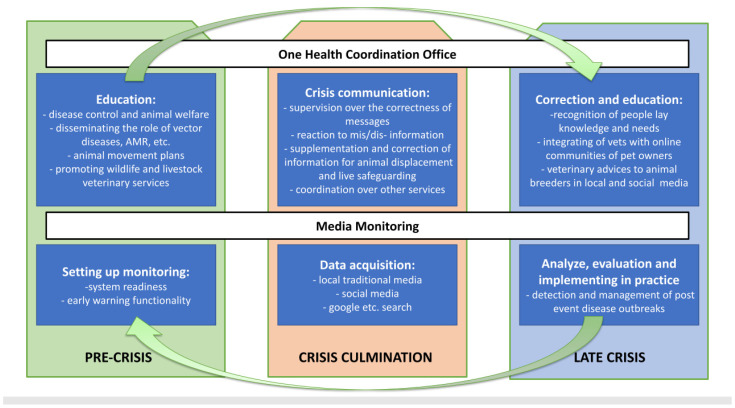
Table suggesting possible interventions in different phases of veterinary crisis caused by disaster. PRE-CRISIS corresponds to situation before a catastrophic event appears and also contains our (i) early signs phase; CRISIS CULMINATION is (ii) acute catastrophic phase; LATE CRISIS contains (iii) beginning of recovery and (iv) disease outbreaks phases.

**Table 1 vetsci-11-00140-t001:** Characteristics of discourse in main animal-related topics.

Topic	Methods of Evaluation	Standard Characteristics	Nontrivial Characteristics
Missing and displaced animals	Media coverage and public discourse analysis; studies on word-of-mouth spread of information.	Focus mainly on missing companion animals shared on social media. Limited discourse on displaced livestock.	Uncommon absence of wild animal discussion compared to discourse on other catastrophes or animal-related issues [7].
Dead animals	Mainstream media agenda-setting studies, content analysis (text and graphics), keyword placement and sentiment analysis.	Articles may not mention animals in text but use emotionally charged images of carcasses. On the other hand, discussion on dead animals often accompanied calculations of economic losses: farmers lost their livestock, equipment (buildings, machinery) and animal feed.	Use of dead animals’ images for emotional impact (clickbait) despite lack of relevant content in mentions. Loss of animal-related accessories (feed and tools) is more important to farmers than livestock.
Infectious diseases and health issues	Community detection (discovery of support groups), topic analysis, measuring intensity of topics and queries on Google and heuristic analysis of time series, including (cross-) correlation of signals in Google and social/local media versus surveillance.	Multiple discussions of epidemiological and epizootiological threats, post-flood topics include human and animal gastroenterological diseases, food/feed contamination and water pollution concerns. Animal owners exchanged information about animal treatment. Many mentions refer to the extent of damage and point out the need for proper management of the situation, given that it is of major public health concern. Tendency of population to seek health information from non-expert sources (possible misinformation).	Broad coverage of health threats, and sophisticated discussions of potential risks (e.g., West Nile fever, leptospirosis which incidences raised after flood or possible re-emerging diseases such as cholera, malaria or dengue fever) not widely acknowledged by general population, which was interested in gastroenterological diseases and acute treatment of their animals only. Mentions about incompetence of authorities are seen in post-acute phases of flood.

## Data Availability

Data (https://zenodo.org/doi/10.5281/zenodo.10451064, accessed on 19 March 2024) and tutorial to use media listening for veterinarians are available at https://github.com/ajarynowski/flood (accessed on 19 March 2024).

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
