# Peer review of "Animal Health Discourse during Ecological Crises in the Media—Lessons Learnt from the Flood in Thessaly from the One Health Perspective"

_vetsci, 2024, doi:10.3390/vetsci11040140_

Round 1

Reviewer 1 Report

Comments and Suggestions for Authors

This paper examines a significant and timely topic using original methodology. I have two minor suggestions for strengthening this paper. In the Summary section and in line 45, the meaning of the word "communication" needs to be clarified. Specify what direction of communication you refer to (e.g., between emergency responders and the public; between veterinarians and responders; or something else). At some points in the paper, I understood you to be referring to communication between veterinary responders and farmers. At other points, I understood you to mean communication with the public via media. It would help to clear this up. 

My second suggestion refers to another point that needs clarification. In line 53, you use the term "precision farming." It would help to have a clause or a sentence explaining what this means. 

As you can see, both of these suggestions are quite easy to remedy. 

Comments on the Quality of English Language

It would be helpful to have the paper edited by a native English speaker. Most of the English is good, but there are a few consistent errors (like definite and indefinite articles) that could easily be cleared up. 

Author Response

Thank you for your constructive feedback. We have clarified the various types of communication discussed in the paper, specifically outlining the contexts in which we refer to communication among different expert groups, authorities, and laypeople. Additionally, we have included a definition and brief explanation of 'precision farming'—an approach to farm management that employs information technology and a broad array of tools, such as control systems, sensors, and software, to monitor and optimize agricultural production processes. These revisions aim to enhance the coherence of the paper and ensure that the concepts are clearly understood. The article was proofread, i.e., the use of definite and indefinite articles was verified.

Reviewer 2 Report

Comments and Suggestions for Authors

The manuscript entitled “Animal health discourse during ecological crisis in the media. Lessons learnt from the flood in Thessaly from the One Health perspective” represents a very interesting and difficult topic to approach.

The problem relating to all matters of veterinary interest in periods of crisis or during non-epidemic emergencies is a current topic that we will have to deal with rigorously at a global level.

The aspects taken into consideration by the Authors are also useful for the purposes of better use of social media with precise awareness-raising purposes, but above all for the organization of a rapid and coordinated intervention action in specific critical situations.

Although the study does not clearly fall into the category of ecological descriptive observational epidemiological studies, it nevertheless represents an interesting source of information that could also be used to propose new organizational strategies in non-epidemic emergency situations.

The manuscript, which as a simple communication does not have excessive claims, would be greatly enriched if the authors tried to put forward hypotheses for the implementation of public veterinary intervention strategies in cases of non-epidemic emergencies with the use of social media and other means. technologies that can also involve private citizens with a view to synergy with the established government system.

I am fully aware that I require a great effort from the Authors, but I would be happy just to see a table of proposals with any indications on the methods for evaluating the effectiveness of these proposals in implementing the work.

Author Response

We acknowledge the importance of integrating more explicit strategies for public veterinary intervention, particularly in non-epidemic emergencies, and the potential of local, new, and social media, as well as technological tools, to facilitate these strategies. In response to your suggestion:

  • We have developed a section dedicated to hypothesizing the implementation of these intervention strategies within an adaptive system (pre-, during, and early post-crisis),
  • Prepared an infographic [Fig. 3] with a visually detailed table of proposals concerning one health management and media monitoring, alongside proposed timely interventions for these issues,
  • Added a column in Table 1 with methods for evaluating the main issues identified in this research.

Our aim is to enrich the manuscript by offering practical insights that could assist in organizing rapid and coordinated response actions in critical situations, as well as fostering a synergistic relationship between private citizens and the ONE public health system. We are aware that our approach deviates from a typical epidemiological study, making the choice of a suitable name challenging. Ecological studies, such as ours, analyze data (like our media indexes) at the population level (in this case, Greek-speaking people), assessing overall frequencies across different subpopulations (topics in our study) and seeking associations. The essence of ecological studies lies in their ability to examine relationships between intensities or sentiments across various topics. It is descriptive (not an analytical study) because we qualitatively examine selected topics.